# Limits to detecting epistasis in the fitness landscape of HIV

**Avik Biswas**[1,2], **Allan Haldane**[1,2], **Ronald M. Levy**[1,2,3]*

**1** Department of Physics, Temple University, Philadelphia, PA, United States of America, **2** Center for Biophysics and Computational Biology, Temple University, Philadelphia, PA, United States of America, **3** Department of Chemistry, Temple University, Philadelphia, PA, United States of America

* ronlevy@temple.edu

**Data Availability Statement:** The code used for model inference and associated utilities is publicly available through Haldane et al., 2020 (https://doi.org/10.1016/j.cpc.2020.107312). HIV protein sequences, including sequence alignments, consensus sequences, etc. are obtained from the

## Abstract

The rapid evolution of HIV is constrained by interactions between mutations which affect viral fitness. In this work, we explore the role of epistasis in determining the mutational fitness landscape of HIV for multiple drug target proteins, including Protease, Reverse Transcriptase, and Integrase. Epistatic interactions between residues modulate the mutation patterns involved in drug resistance, with unambiguous signatures of epistasis best seen in the comparison of the Potts model predicted and experimental HIV sequence "prevalences" expressed as higher-order marginals (beyond triplets) of the sequence probability distribution. In contrast, experimental measures of fitness such as viral replicative capacities generally probe fitness effects of point mutations in a single background, providing weak evidence for epistasis in viral systems. The detectable effects of epistasis are obscured by higher evolutionary conservation at sites. While double mutant cycles in principle, provide one of the best ways to probe epistatic interactions experimentally without reference to a particular background, we show that the analysis is complicated by the small dynamic range of measurements. Overall, we show that global pairwise interaction Potts models are necessary for predicting the mutational landscape of viral proteins.

## Introduction

A major challenge in biological research, clinical medicine, and biotechnology is how to decipher and exploit the effects of mutations [1]. In efforts ranging from the identification of genetic variations underlying disease-causing mutations, to the understanding of the genotype-phenotype mapping, to development of modified proteins with useful properties, there is a need to rapidly assess the functional effects of mutations. Experimental techniques to assess the effect of multiple mutations on phenotype have been effective [2–5], but functional assays to test all possible combinations are not possible due to the vast size of the mutational landscape. Recent advances in biotechnology have enabled deep mutational scans [6] and multiplexed assays [7] for a more complete description of the mutational landscapes of a few proteins, but remain resource intensive and limited in scalability. The measured phenotypes depend on the type of experiment and are susceptible to changes in experimental conditions

Stanford HIV drug resistance database (HIVDB, https://hivdb.stanford.edu) and the Los Alamos HIV sequence database (https://www.hiv.lanl.gov/content/sequence/HIV/mainpage.html). The details, filtering criteria (if any) are mentioned in the Materials and Methods section. The protein sequences for the molecular clones NL4-3 and HXB2 are obtained from GenBank (Clar et al., 2015, https://doi.org/10.1093/nar/gkv1276) with accession number AF324493.2 and K03455.1, respectively. A repository containing the final MSAs used in the study is available at https://github.com/ComputationalBiophysicsCollaborative/HIV_MSAs. Drug resistance information, including list of drug-resistance-associated mutations are obtained from the Stanford HIVDB and from Wensing et al., 2019 (https://www.ncbi.nlm.nih.gov/pmc/articles/PMC6892618/). Any additional information can be obtained on request to the corresponding author.

**Funding:** This work has been supported by the National Institutes of Health through grants awarded to RML (U54-AI150472, R35-GM132090, S10OD020095). The National Science Foundation also provided funding through a grant awarded to to RML and AH (1934848). The funders had no role in study design, data collection and analysis, decision to publish, or preparation of the manuscript.

**Competing interests:** The authors have declared that no competing interests exist.

making the comparison between measurements difficult [8]. These methodologies are also utilized under externally applied conditions, but how *in vitro* selection pressures can be extended to the interpretation of pressures *in vivo* is not always clear [9].

Potts sequence covariation models have been developed for the identification of spatial contacts in proteins from sequence data [10–19] by exploiting the wealth of information available in protein sequences observed in nature, and have also been successfully used to infer the fitness landscape and study mutational outcomes in a wide variety of protein families in viruses to humans [1, 20–29]. The Potts model is a generative, global pairwise interaction model that induces correlations between residues to all orders, such as triplet and quadruplet correlations. Given a multiple sequence alignment (MSA) of related protein sequences, the Potts probabilistic model of the network of interacting protein residues can be inferred from the pair correlations encoded in the MSA, and can be used to assign scores to individual protein sequences. The extent to which sequence scores correlate with experimental measures of fitness can then be analyzed. The context dependence of a mutation, termed "epistasis", determines the favorability/disfavorability of the mutation in a given genomic sequence background, and the Potts model predictions can be used to predict the likelihoods of mutations in a variety of sequence backgrounds.

The HIV pandemic is the result of a large, genetically diverse, and dynamic viral population characterized by a highly mutable genome that renders efforts to design a universal vaccine a significant challenge [30] and drives the emergence of drug-resistant variants upon antiretroviral (ARV) therapy. Gaining a comprehensive understanding of the mutational tolerance, and the role of epistatic interactions in the fitness landscape of HIV is important for the identification and understanding of mutational routes of pathogen escape and resistance.

In this work, we explore the limits to detecting epistasis and the role of epistatic interactions between sites in modulating the fitness landscape of HIV with many mutations, focusing on the drug target proteins, protease (PR), reverse transcriptase (RT), and integrase (IN), as well as the emerging target protein of capsid (CA). We first show that the evidence for long-range epistasis is strong (in HIV) based on the analysis of the high-order marginals of the MSA distribution (up to subsequences of length 14). The observed effects of epistasis in determining the higher-order mutational patterns, however, differs significantly between drug-resistance-associated residues, and non-drug-resistance-associated sites due to the higher "evolutionary conservation" at sites not associated with drug resistance. In contrast to the effects of epistasis on higher-order marginals, we find, in accordance with the current literature [31], that the evidence for epistasis from experimental measures of HIV fitness, such as viral replicative capacities, is weak; as both a correlated Potts and a site-independent model (devoid of interactions between sites) can capture replicative capacities almost equally well. This is primarily because experimental fitness measurements are generally carried out for single-point (or few-point) mutations in specific laboratory molecular clones, that are close in sequence to the consensus sequence. Instead, in the comparison of higher-order marginals, unambiguous signatures of epistasis are observed. Although double mutant cycle experiments in principle provide the classic, biophysical way to examine epistasis, we demonstrate with numerical examples that accurate predictions of double mutant cycles are difficult due to the small dynamic range of the measurements making them much more susceptible to noise. While fitness measures such as thermostability, activity, or binding energetics, etc. of a protein generally do not all contribute to fitness in the same way, we further find that the Potts model provides a more general representation of the protein fitness landscape capturing contributions from different features of the landscape, replicative capacities and folding energetics, that are not fully captured by either measurement on their own.

## Results and discussion

Protein sequence covariation models have been extensively used to study networks of interacting residues for inference of protein structure and function. The Potts model is a maximumentropy model based on the observed mutational correlations in a multiple sequence alignment (MSA) and constrained to accurately capture the bivariate (pairwise) residue frequencies in the MSA. A central quantity known as the "statistical" energy of a sequence $E(S)$ (Eq 2, Methods) is commonly interpreted to be proportional to fitness; the model predicts that sequences will appear in the dataset with probability $P(S) \propto e^{-E(S)}$, such that sequences with favorable statistical energies are more prevalent in the MSA. $P(S)$ describes the "prevalence" landscape of a protein and the marginals of $P(S)$ can be compared with observed frequencies in a multiple sequence alignment. Previous studies have indicated that the Potts model is an accurate predictor of "prevalence" in HIV proteins [20, 21, 23, 32–36]; "prevalence" is often used as a proxy for "fitness" with covariation models serving as a natural extension for measures of "fitness" based on experiments and model predictions have been compared to different experimental measures of "fitness" with varying degrees of success [1, 21, 23, 28, 32, 34, 36]. Site-independent models, devoid of interactions between sites have also been reported to capture experimentally measured fitness well, in particular for viral proteins [1, 31] with studies (on HIV Nef and protease) implying that the dominant contribution to the Potts model predicted sequence statistical energy comes from site-wise "field" parameters $h_i$ (see Methods) in the model [28, 36]. In this study, we show that interaction between sites is necessary to capture the higher order (beyond pairwise) mutational landscape of HIV proteins for functionally relevant sites, such as those involved in engendering drug resistance, and cannot be predicted by a site-independent model. The correspondence between model predictions of fitness based on "prevalences" in natural sequences with experimentally measured "fitness" is however, confounded by a number of different factors. Here, we explore comparisons between model predictions and "fitness" experiments (Fig 1) focusing primarily on three HIV enzymes: Protease

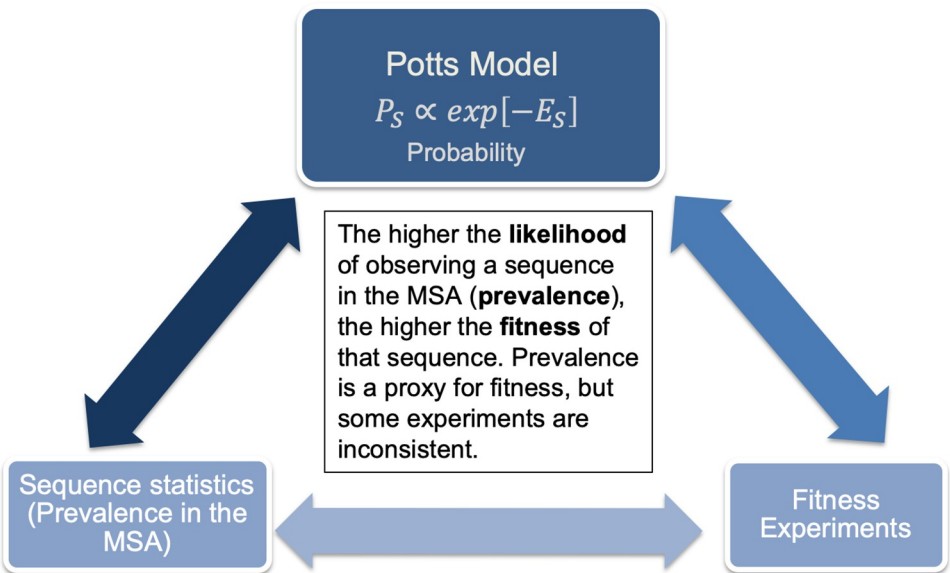

**Fig 1. The correspondence between sequence covariation models and sequence statistics in multiple sequence alignments is very strong across different HIV proteins.** The correspondence between either covariation models, or "prevalences" in multiple sequence alignments, with other experimental measures of "fitness" is less clear and often inconsistent between different statistics and measures of fitness.

(PR), Reverse Transcriptase (RT), and Integrase (IN) that have been targets of antiretroviral therapy (ART) over the past several decades, as well as viral Capsid (CA), which is fast emerging as a promising new target for drug therapy.

## "Prevalence" landscape of HIV proteins and the role of correlations between residues

An important statistic of the multiple sequence alignment is the sequence diversity and the level of conservation in the protein or protein family which is represented in the distribution of the number of mutations in the constituent sequences. Fig 2 shows the distribution of the number of mutations (hamming distances) from the HIV-1 subtype B wild-type consensus sequence in MSAs containing drug-experienced HIV-1 sequences, and distributions predicted by the Potts and independent models. The Potts model predicts a distribution of mutations that closely represents the dataset distribution, whereas the independent model predicts a distribution that differs especially near the the ends of the distribution where the number of mutations is either very low or very high. This provides support for the importance of epistasis in these datasets. However, in Fig 9 of S1 File we also show that for some datasets the difference between the Potts and Independent distributions is small, and so may be a less reliable test of the importance of epistasis. The importance of pairwise interactions is also apparent through the fact that the Potts model also accurately predicts the likelihoods and "entrenchment" of mutations based on the sequence background, as has been verified using aggregate sequence statistics from the MSA [35].

But the most direct and strongest evidence of the ability of the Potts model to capture epistatic interactions is seen in its ability to reproduce the higher-order marginals of the MSA, upto order 14 in Fig 3, much beyond the pairwise marginals which the model is parameterized to capture. While the prevalence of sequence marginals (subsequence frequencies) can be compared directly with Potts model predictions, this is not possible for predictions for complete sequence probabilities because most sequences in an unbiased MSA are observed only once due to the minuscule sample size in comparison to the vast size of sequence space. Only sequence marginals up to sizes $\sim 14$ residues, depending upon protein family, are observed

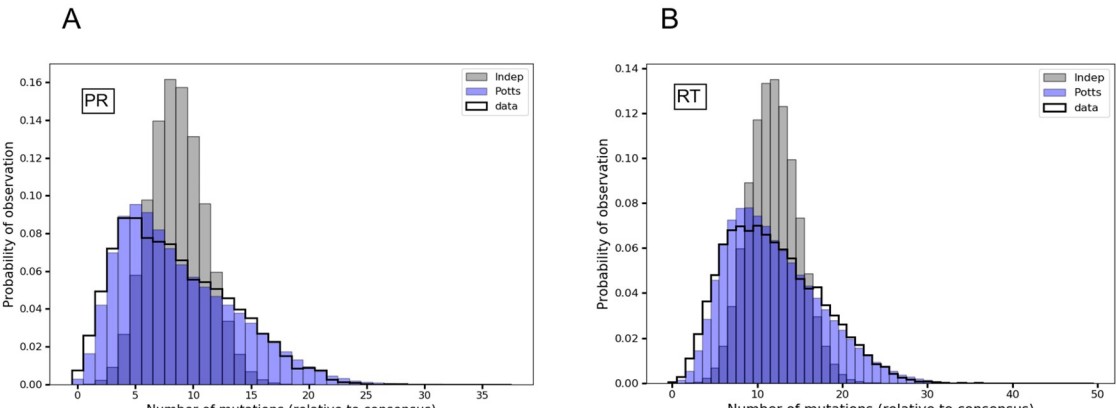

**Fig 2. Distribution of the number of mutations (hamming distances) in drug-experienced HIV-1 sequences as captured by the Potts and independent models.** Probabilities of observing sequences with any $k$ number of mutations relative to the HIV-1 subtype B wild-type consensus sequence as observed in original MSA (black) and predicted by the Potts (blue) and independent (gray) models are shown for HIV-1 protease (PR) in (A), and reverse transcriptase (RT) in (B), respectively. The independent model predicted distribution does not accurately capture the distribution of hamming distances in the dataset MSA, especially near the ends of the distribution with either very low or very high number of mutations, where the epistatic effects can be more significant.

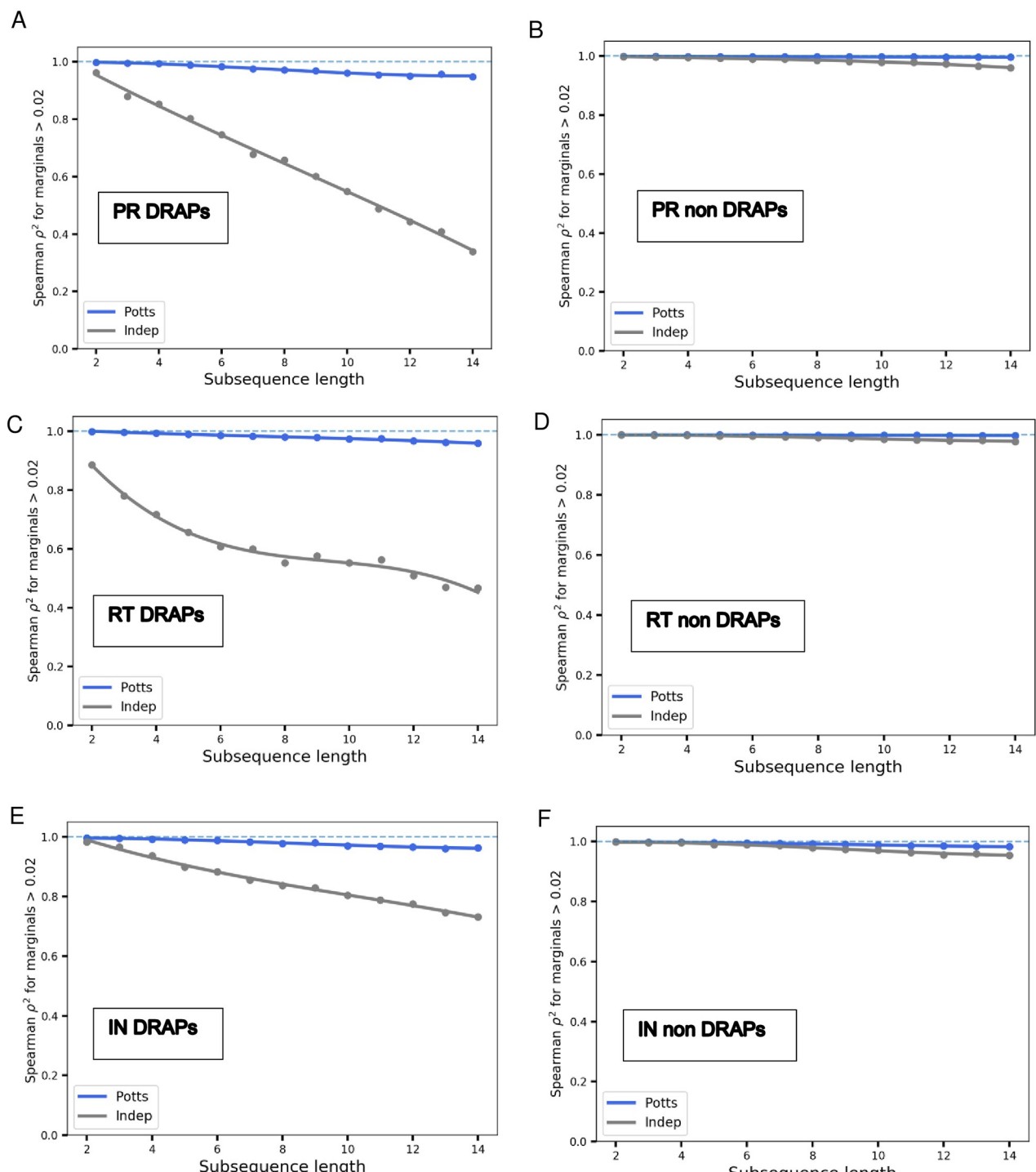

**Fig 3. Potts model is predictive of higher-order marginals in the sequence MSA.** For each subsequence of length 2 to 14, marginal frequencies are determined by counting the occurrences in the MSA and computed for 500 randomly picked subsequences. They are compared with the corresponding predictions of marginal probabilities by the Potts model (blue) and a site-independent model (gray). The Spearman $\rho^2$ between the dataset marginal frequencies and the Potts and independent model predictions for all marginal frequencies above 2% are shown for subsequences picked at random from different combinations of 36 Protease-inhibitor or PI-associated positions in PR (A), 24 Nucleoside-reverse-transcriptase-inhibitor or NRTI-associated positions in RT (C), and 31 Integrase-strand-transfer-inhibitor or INSTI-associated positions in IN (E). Shown in (B), (D), (F), are the same but the subsequences are picked at random from non resistance-associated sites in PR, RT, and IN, respectively. The blue dashed line represents perfect correlation of $\rho^2 = 1$. In all, the Potts model accurately captures the higher-order marginals in the dataset; the independent model however gets progressively worse in capturing the higher-order marginals for resistance-associated sites in (A), (C), and (E). The

role of epistatic interactions is strongly manifested in the effect on drug-resistance-associated positions (DRAPs) (indicating the strong role of correlations at functional positions within the protein). For residue positions not associated with drug resistance, epistatic interactions between sites appear to play a less important role and the site-independent model is sufficient to model the higher-order marginals in the MSA.

with sufficient frequencies such that their marginal counts are a good proxy for the marginal probabilities predicted by the Potts model. Fig 3 shows the rank-correlation between model predicted marginal probabilities and marginal frequencies in the MSA for subsequences of lengths 2–14, with a subsequence being the concatenation of amino acid characters from an often nonconsecutive subset of residue positions. With further increase in the subsequence length, data limitations due to finite sampling become more prominent and the observed and model predicted marginals become dominated by noise. The Potts model's ability to predict higher-order marginals, much beyond the pairwise, for drug-resistance associated sites, while the independent model cannot, provides the most direct evidence of the ability of the Potts model to accurately capture the long-range epistatic interactions that modulate the "prevalence" of amino acid residues at connected sites in the protein. The Potts model is able to accurately predict the higher-order marginal frequencies (which have not been directly fit) at drug-associated sites with a Spearman $\rho^2 \approx 0.95$ for the longest subsequence (of length 14) in PR, whereas, the correlation for the independent model deteriorates sharply with subsequence length (Fig 3A, 3C and 3E) with a Spearman $\rho^2 \approx 0.34$ for the longest subsequence in PR.

The strongly interacting nature of the sites in HIV that are involved in engendering drug resistance, is also evident from Fig 3A, 3C and 3E, where the role of epistatic interactions between residues is more pronounced and the site-independent model is not able to capture the higher-order marginals. In contrast, for residue positions that are not associated with drug resistance, the site-independent model can sufficiently recover the higher order marginals in the MSA. Sites in the protein associated with drug resistance, also however, exhibit considerably more variability contributing to their higher site-entropies (Fig 2A of S1 File). The lack of variability at sites can obscure the effect of correlations. To test for this, we selected protease-inhibitor associated and non-associated sites with site-entropy distributions similar to that of the drug-resistance associated sites (Fig 2B of S1 File) and compared their higher order marginals as predicted by the Potts and site-independent models (Fig 2C and 2D of S1 File, respectively). When marginals are chosen from non-drug associated positions with site entropies more similar to those of the drug-associated positions, the role of correlations is more apparent. This is suggestive of strong couplings between sites that are likely to co-mutate, allow for mutations at lesser costs to fitness than the individual mutations alone, resulting in mutational pathways selected for pathogen escape. Such sets of sites are more likely to be associated with resistance, as resistance cannot be achieved through selectively neutral mutations at single sites, in which case drug treatment would likely be ineffective [37].

In contrast to Fig 3A for HIV PR, Fig 3E shows the somewhat improved predictive capacity of a site-independent model in capturing the higher order sequence statistics for drug-resistance associated positions in HIV IN. This is indicative that correlations between drug-resistance-associated sites appear to play a stronger role in protease than in IN. This is also in line with the fact that the IN enzyme is more conserved than PR (Fig 1 of S1 File). Amongst the three drug-target proteins, PR, RT, and IN, the degree of "evolutionary conservation" is considerably higher in IN than in the others. The lack of variability at sites or considerably smaller site-entropies in IN plays a role in obscuring the effect of correlations, as discussed. Furthermore, the MSA depth for IN is also considerably lower than in PR or RT, which adversely affects the quality of the Potts model fit, further making the correlated model less distinguishable from a site-independent one [38].

The majority of the literature on HIV discusses drug resistance in relation to correlated mutations limited to primary/accessory pairs. Fig 3 depicts the effect of correlated mutations on the "prevalence" landscape of HIV well beyond pairwise interactions, upto the 14th order, that is captured accurately by the Potts model. This illustrates the existence of correlated networks of long-range interactions between sites in HIV, which play an important role in determining its evolutionary fitness landscape. The entrenchment of primary resistance mutations in HIV was shown to be contingent on the presence of specific patterns of background mutations beyond the well studied primary/accessory compensatory pairs, and could not be predicted on the basis of the number of background mutations alone [35], also indicating that long-range correlations involving many sites can potentially shape the evolutionary trajectory of the virus.

## From sequence covariation to "fitness"

The Potts model predicted statistical energies $E(S)$ have been established to be a good indicator of the likelihoods ($P(S) \propto e^{-E(S)}$) or "prevalence" of natural sequences in multiple sequence alignments; prevalence has often been characterized as a proxy for fitness with sequences more prevalent in the MSA likely to have a fitness advantage over others. But depending on context, the notion of fitness can entail a variety of experimental measures from replicative capacity (RC), to protein stability, catalytic efficiency, molecular recognition, drug-resistance values, etc., each of which may capture different features of the fitness landscape, and can have varying degrees of correspondence to observed likelihoods in MSAs of natural sequences. In this section, we explore the correspondence between measures of fitness based on experimental replicative capacities of HIV mutants and the Potts model predicted likelihoods in an MSA. The correspondence is confounded by a number of factors such as the reproducibility of experiments, the quality of inferred Potts models, the degree of evolutionary conservation in the proteins amongst others.

Fig 4 shows the correlation between model predicted likelihoods of HIV mutant proteins and measures of fitness based on/related to replicative capacities for four HIV proteins, PR, RT, IN, and p24 CA. The independent model generally performs on par or marginally worse than the Potts model in capturing experimental replicative capacity measurements. Although the difference is somewhat larger for measurements focusing on only drug-resistance-associated mutations indicated with "D" in Fig 4 rather than random mutations or mutations at non-resistance-associated positions indicated with "R", along the lines of Fig 3 for marginal statistics, the difference is not as clear as for marginal statistics. It has been suggested that the independent model performs on par with correlated Potts or advanced machine learning models in capturing experimental fitness measurements for viral proteins, possibly as a consequence of limited diversity of the sequence alignments or, due to a discrepancy between the proxy for viral fitness in the laboratory and the *in vivo* fitness of the virus [1, 9]. Overall, we find that the evidence for epistasis from measures of fitness based on experimental replicative capacities is much weaker compared to that available from the higher-order marginal statistics.

The Potts model is affected by the degree of conservation in the respective proteins which can, not only affect the quality of the model as reflected in the signal-to-noise ratio or SNR (see Methods), but also obscure the effect of correlations between sites. To check how the Potts model predictions may be affected by the quality and sample sizes of the underlying multiple sequence alignments, we look at the effect of using Potts models built on MSAs that all have the same depth but contain different sequences (randomly subsampled from a larger dataset) on the correspondence between experimental and model predictions of fitness (Fig 5). The

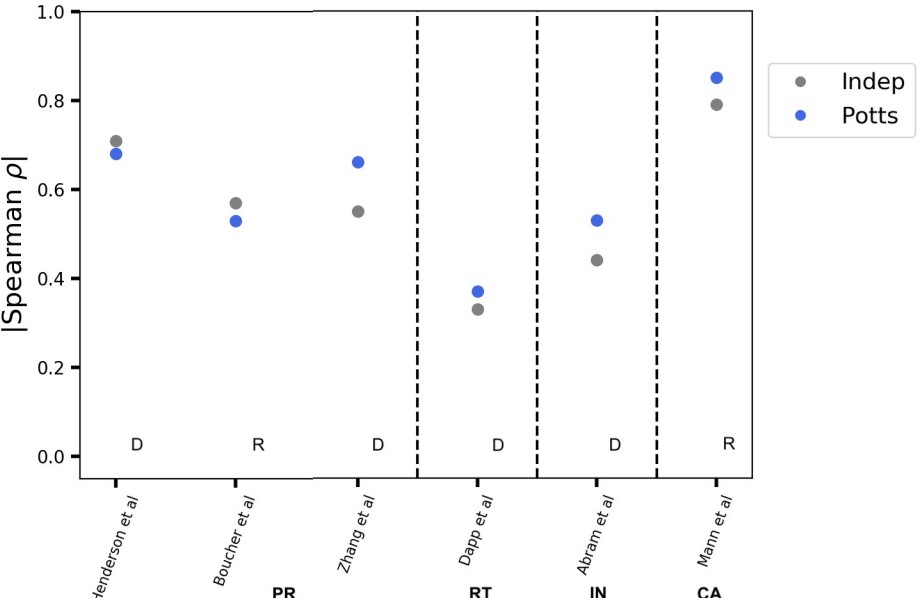

**Fig 4. Survey of correlation between sequence-based predictions and experimental measures of "fitness" based on replicative capacity.** Spearman correlation coefficients, $\rho$ between prevalence-based measures of fitness as predicted by the Potts (blue) and independent (gray) models and experimental measurements related to replicative capacities are shown across four different HIV proteins: PR, RT, IN, and p24 CA. Experimental data are obtained from [3, 21, 28, 39–41]. Experiments reporting fitness measurements for random mutations are marked with an "R" and experiments reporting drug-resistance only mutations are marked with a "D". Correlation is not consistent between different experiments for the same protein. The Potts model generally (marginally) outperforms the independent model in capturing experimentally measured replicative capacities or measures related to replicative capacities.

correlation also decreases slightly because MSAs of depth half that of the original (reference Potts) are used for mutational fitness predictions. Overall, this gives an estimate that the statistical error associated with Potts model predictions of fitness is low. The correspondence between the predicted fitness based on Potts prevalence and on experiment also depends in part on which experimental assays are chosen as a proxy for fitness and the extent to which they can capture phenotypes that are under direct, long-term selection [42], as illustrated in Figs 3 and 4 of S1 File. Fig 3 of S1 File shows little correlation between two closely related experimental measures of fitness for HIV PR; one based on replicative capacity [5] and the other based on selection coefficients [40]. Interestingly, the Potts model predictions correlate well with one of the datasets. More careful analysis is needed to improve our understanding of which experimental measures contribute most to the "prevalence" landscape captured by Potts models.

**Effect of epistasis on measurements of fitness and double mutant cycles in HIV.** Double mutant cycles provide a biophysical means to interrogate epistatis without reference to a specific sequence background [43]. For a pair of mutations $\alpha$, $\beta$ at positions $i$, $j$ in the protein respectively, the strength of epistatic interactions can be quantified using the difference between the sum of the independent mutational effects, $\Delta E_\alpha^i + \Delta E_\beta^j$, and the effect of the corresponding double mutation, $\Delta E_{\alpha\beta}^{ij}$.

$$\Delta\Delta E_{\alpha\beta}^{ij} = \Delta E_{\alpha\beta}^{ij} - (\Delta E_\alpha^i + \Delta E_\beta^j) \tag{1}$$

If $\Delta\Delta E \neq 0$, then the two mutations are epistatically coupled, whereas if $\Delta\Delta E = 0$, then the mutations are mutually independent. As fitness is inversely proportional to the Potts energy,

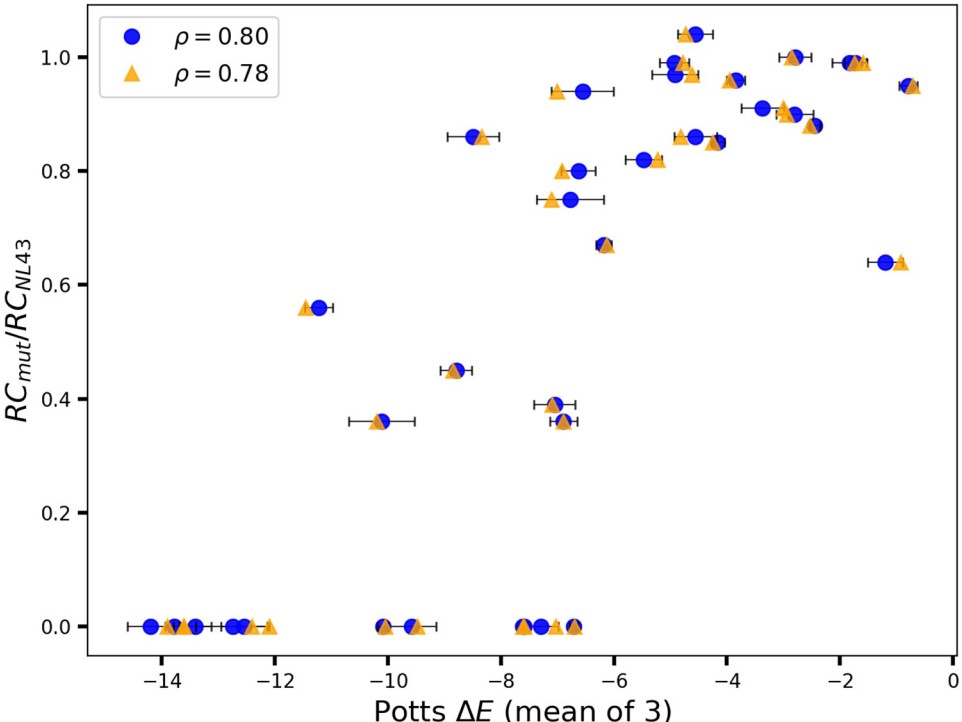

**Fig 5. Error estimate in Potts model predictions of fitness.** Figure shows replicative capacity based experimental fitness measurements (from [21]) compared to Potts model predicted likelihoods of mutations in HIV-1 CA. The Potts model predicted values shown in "blue circles" correspond to the mean of 3 predictions based on jackknife tests with error bars indicating the standard deviations. Random values are then picked from within each standard deviation to represent each Potts model prediction (shown as "orange triangles") and the corresponding effect on the correlation coefficient is observed. Spearman rank-order correlation $\rho = 0.8$ for mean of 3 predictions, and $\rho = 0.78$ for random selection of data from within the margin of error. For jackknife tests, three sets of $\approx 1024$ weighted patient sequences are subsampled at random from the original MSA of $\approx 2200$ weighted sequences, and new Potts models are then inferred based on each set. For comparison, the Spearman rank-order correlation is $\rho = 0.85$ for the original Potts model (based on an MSA of 2200 sequences) predictions compared to experimental values (Fig 5A of S1 File). Figure shows an estimate of the error associated with Potts model predictions of likelihoods of mutants stemming from sampling of sequences in the MSA and its effect on the correspondence with experimental measures of fitness.

$\Delta\Delta E > 0$ implies that the mutations are beneficial/co-operative to each other and *vice versa*. The dynamic range of double mutant cycles is an order-of-magnitude smaller than the predictions/measurements of likelihoods/fitness effects of mutations ($\Delta E$s), shown in Fig 6A. Double mutant cycle measurements/predictions ($\Delta\Delta E$s) are therefore, much more susceptible to noise, and strongly affected by both the quality of the experimental measurements, as well as finite sampling errors that affect the Potts model fit, making accurate numerical predictions very difficult. The MSA depth also plays a role in degrading the quality of the Potts model double mutant cycle predictions, $\Delta\Delta E$s much faster than the fitness effect of point mutations, $\Delta E$s (Fig 6 of S1 File). The sensitivity and possible interpretation of experimental measurements for very detrimental mutational changes is crucial for accurate prediction of double mutant cycles. When experimental replicative capacities for example, of a single and a double mutant(s) are both zero (the virus is dead), there is no comparative experimental data to inform if and which mutation(s) are more deleterious. In contrast, the calculated likelihoods from the Potts model are quantifiable for both.

The correspondence between the $\Delta\Delta E$ values predicted by the Potts model and the equivalent experimental values would provide a strong confirmation of epistasis that can be directly

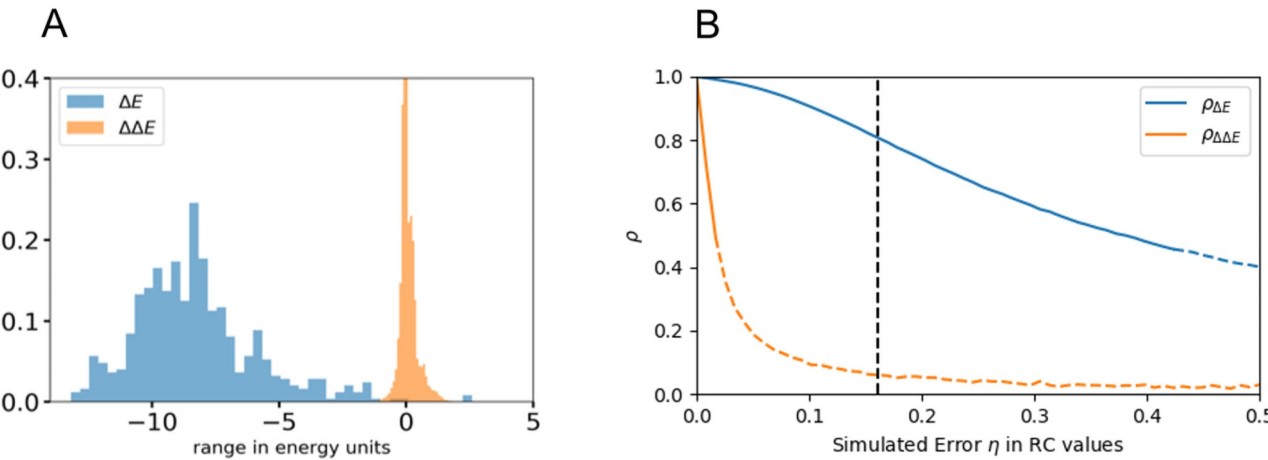

**Fig 6.** (A) The dynamic range of the measurements (experimental) or predictions (model) of the epistatic effects through the use of double mutant cycles is an order of magnitude smaller than the range of measurements/predictions of the fitness/likelihoods of point mutations. This makes predictions for double mutant cycles more susceptible to noise. (B) Simulation of the expected correlation of the Potts model prediction to experimental values for $\Delta E$ and $\Delta\Delta E$ as a function of simulated experimental noise $\eta$, showing that the the correlation for $\Delta\Delta E$ drops much more quickly. The dotted section of the curves show where the *p*-value for the $\Delta\Delta E$ correlation is >0.05, or insignificant, showing that noise can make it impossible to verify $\Delta\Delta E$ values even when $\Delta E$ values are well predicted. The level of noise corresponding to $\Delta E$ correlation of $\rho \approx 0.8$, as in Fig 4 column 6 for Capsid, is shown in dashed black.

experimentally measured; but in practice, such a comparison is often statistically not possible due to experimental and statistical uncertainties(s). In Fig 6B, we illustrate how error in individual fitness measurements can cause the double mutant cycle predictions to be unverifiable even when there exists good correspondence between Potts model and experimental fitness predictions. In this simulated test, the Potts model $\Delta E$ predictions for capsid (for mutation datapoints shown in Fig 4 (column 6) are rescaled to have the same range and scale as experimental replicative capacity values, and are used as simulated replicative capacity values. Varying amounts of random noise representing experimental error(s) and modelled as Gaussian white noise with mean 0 and standard deviation $\eta$ are added to each $\Delta E$ value, which are then interpreted as simulated experimental RC values. The simulated RC values are taken to be the "ground truth" which are used to evaluate double mutant cycles, to compare to the Potts predictions. The Spearman rank-order correlation coefficients between the Potts model predicted and simulated experimental RC values, as well as double mutant cycle values, are then computed for the mutation residue-identities as available in our experimental dataset, and the process is repeated for varying degrees of noise strength (specified by varying $\eta$), representing varying degrees of experimental uncertainty. The correlation between model predicted and simulated experimental RC values are shown in Fig 6B. We see that even when the $\Delta E$ correlation with the simulated RC is as high as $\sim 0.8$ (as is observed for capsid), the corresponding $\Delta\Delta E$ correlation with differences in Replicative Capacity between double mutants and the corresponding sum of single mutants is very low, $\sim 0.1$ and is typically not statistically significant. For a correlation between model and experimental RC values $\sim 0.6$ as observed for HIV protease (Fig 4 column 3), the same result is obtained, namely double mutant cycle analysis can not be used to verify epistatic interactions for HIV protease (Fig 7A of S1 File). Indeed, the correlation with double mutant cycles computed from the experimental values in Fig 4 is very low and statistically insignificant (Fig 7B of S1 File) in agreement with this test. Nevertheless, many of the strongest predicted (by the Potts model) double mutant cycles in HIV proteins, indeed qualitatively agree with the effects studied in the literature, especially amongst those involving compensatory pairs of drug-resistance mutations in HIV drug-target proteins (S2A and S2B Fig, S2A and S2B of S2 File).

**Contribution of the changes in structural stability due to a mutation to the predicted likelihoods of mutant sequences.** In this section, we explore the contribution from changes in structural stability due to a mutation to its Potts model predicted likelihood(s). To explore the impact of a mutation on structural stability, we employ a well-known protein design algorithm called FoldX [44, 45], which uses an empirical force field to determine the energetic effects of a point mutation. FoldX mutates protein side chains using a probability-based rotamer library while exploring alternative conformations of the surrounding side chains, in order to model the energetic effects of a mutation. We observe good correspondence between Potts model predicted likelihoods and FoldX predicted changes in structural stabilities of mutations in Fig 7B, and Fig 8B of S1 File for a set of multiple inhibitor-associated mutations (from [28]) in PR. There also exists statistically significant correlation between experimentally measured replicative capacities of these mutations and their Potts model predicted likelihoods (Fig 7A, and Fig 8A of S1 File), but the FoldX predicted changes in structural stabilities do not correlate so well with experimentally measured replicative capacities (Fig 7C, Fig 8C of S1 File). This is indicative that different measures of fitness such as thermostability, activity, or folding energetics of a protein do not generally contribute to fitness in the same way [1]. While some measures or properties being tested may only have an indirect context-dependent impact on fitness, "prevalence" in multiple sequence alignments of thousands of protein sequences may be more reflective of the overall survival fitness. Fig 7 and Fig 8 of S1 File show that the Potts model can capture contributions to fitness from both structural stabilities measured by FoldX, and from other aspects of the viral replicative life-cycle measured by replicative capacity experiments, which are not captured completely by either measurement on its own.

## Conclusion

Fitness is a complex concept at the foundation of ecology and evolution. The measures of fitness range from those such as replicative capacity, protein stability, catalytic efficiency, that can be determined experimentally in the lab to measures stemming from the "prevalence" in collections of sequences obtained from nature, that can be quantified and compared using predictions of coevolutionary models which encode mutational patterns in multiple sequence alignments. For viral fitness measurements, the large majority of studies focus on measures like selection coefficients or replicative fitness within hosts or cells in culture. Potts models of sequence co-variation provide a measure of fitness tied to the frequency of sequences appearing after longer *in vivo* evolutionary times in the virus' natural environment.

The functions of proteins are defined by the collective interactions of many residues, and yet many statistical models of biological sequences consider sites nearly independently [31]. While studies [1] have demonstrated the benefits of including interactions to capture pairwise covariation in successfully predicting the effects of mutations across a variety of protein families and high-throughput experiments, for viral proteins, the predictions of mutational fitness by pairwise or latent-space models often fall short of predictions by site-independent models. It has been suggested that this could be a consequence of the limited diversity of the sequence alignments [1] or due to a discrepancy between the proxy for viral fitness in the laboratory and the *in vivo* fitness of the virus [9]. Here, we show that the signatures of epistasis are best manifested for viruses like HIV in the comparison of the Potts model predicted and experimental HIV sequence "prevalences" when expressed as higher-order marginals of the sequence probability distribution. *Gupta and Adami* have shown that epistasis in HIV can also be detected using a different metric, the pairwise mutual information [46]. The approach presented here goes further and accurately describes the higher order mutational landscape of the virus in response to external selection pressures such as drug exposure.

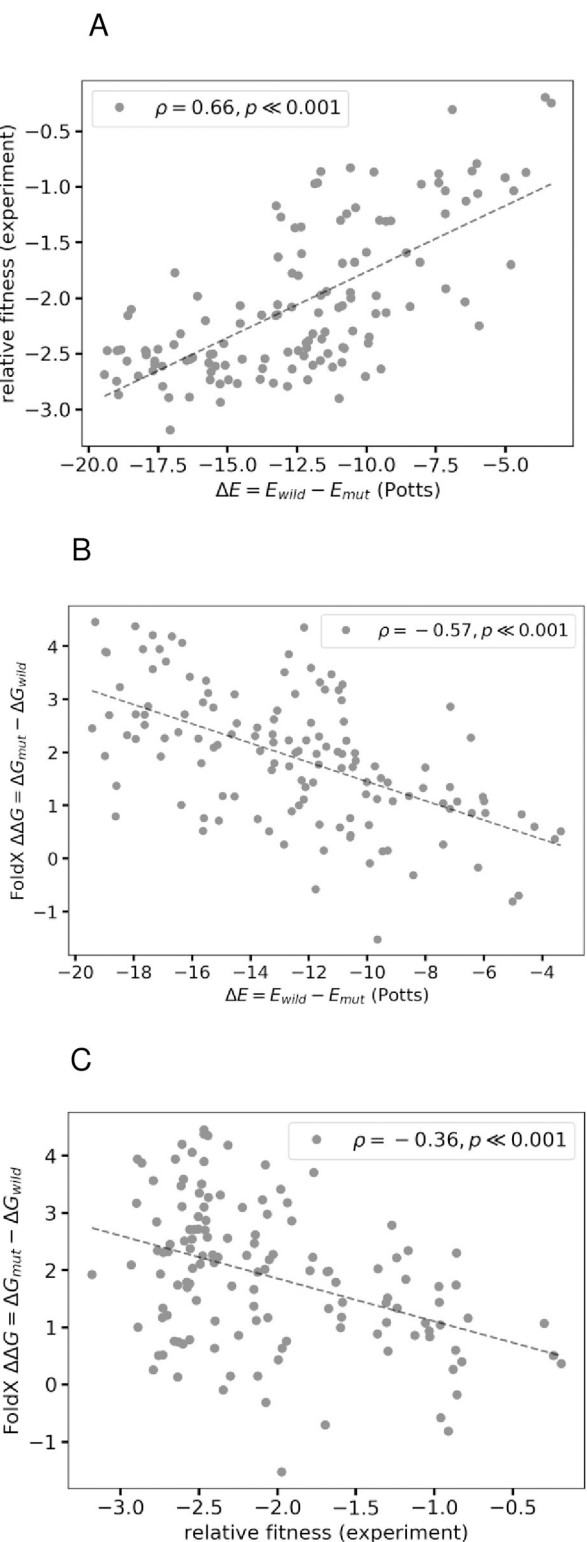

**Fig 7. Potts model captures different features of the fitness landscape.** Figure shows that the Potts model predicted ΔEs can capture different features of the fitness landscape that may be orthogonal, and may not correlate well with each other. (A) Relative fitness (replicative capacity) measurements obtained from deep mutational scanning of HIV-1 variants [28] involving combinations (of three or lesser) of mutations in protease associated with resistance to (particularly second-generation) inhibitors in clinic, are compared to changes in Potts statistical energies, ΔEs with a

Spearman rank-order correlation, $\rho = 0.66$ ($p \ll 0.001$). [28] also report statistically significant correlation ($|\rho| = 0.46$) with a Potts model inferred using the Adaptive Cluster Expansion (ACE) algorithm. (B) FoldX predicted changes in folding energies, $\Delta\Delta G$s (PDB: 3S85) of the mutations also correlate well with Potts predicted changes in statistical energies, $\Delta E$s for the same (Spearman $\rho = -0.57$). The HIV-1 protease structure (PDB: 3S85) is used as reference, repaired using the RepairPDB function in the FoldX suite, and the free energy of mutants is calculated with the BuildModel function under default parameters. Changes in structural stability due to mutations correlate well with their predicted likelihoods (estimated by the Potts model $\Delta E$s) as seen here with a Spearman rank-order correlation, $\rho = -0.57$ ($p < 0.001$) between the two. However, FoldX calculations are susceptible to small changes in structure that can be caused by the presence of small-molecule ligands, etc. For another PDB:4LL3, we still find statistically significant correlation between the two ($\rho = -0.64$). (C) Experimental relative fitness measurements however, do not correlate as well with FoldX predicted changes in folding energies due to the mutations ($\rho = -0.36$).

The model, which is parameterized to reproduce the bivariate marginals in the MSA, also accurately captures the higher order marginal probabilities (seen in Fig 3A, 3C and 3E, upto the 14th order) in the MSA for sets of drug-resistance associated positions; whereas, the fidelity of a site-independent model decreases much more rapidly with the size of the marginal. We further show that epistatic interactions are particularly important in determining the higher order mutation patterns of drug-resistance-associated sites in HIV; in clear contrast with non-drug-resistance-associated positions, as the virus evolves under drug pressure employing the most strongly interacting positions in mutational pathways.

It has been suggested that the success of models based on sequence covariation at recapitulating high-throughput mutation experiments depends in part on the extent to which experimental assays can capture phenotypes that are under direct, long-term selection [1]. For some proteins, such as nonessential peripheral enzymes or signaling proteins, the property being tested in the laboratory may only have an indirect, context-dependent impact on the organism. We observe higher correlation between Potts model and experiment for the structural protein Capsid than other enzymatic proteins like PR, RT, or IN. A similar correlation has been observed for Capsids of other types of viruses [47]. This indicates that changes in CA perhaps have a more direct effect on the viral lifecycle than enzymatic proteins. However, the evidence for epistasis from fitness measurements based on replicative capacity experiments remains weak as both the Potts and independent models often show comparable degrees of correlation with experiment, and the distinction may not be statistically significant. While double mutant cycles provide a well established biophysical way to probe epistatic effects without reference to a particular sequence background, the order-of-magnitude smaller dynamic range makes accurate quantitative predictions very difficult and we only see weak evidence for epistasis through double mutant cycles.

Different measures may also contribute to fitness in different ways. In this study, we employ FoldX to probe the contribution of structural changes and folding energetics due to mutations to their predicted/observed likelihoods, finding that the Potts model predicted likelihoods of mutations in HIV correlate well with FoldX predicted changes in free energies. FoldX predictions, however, do not correlate well with experimental replicative capacity measurements. This is suggestive that the overall fitness landscape predicted by the Potts model includes contributions from many different features, some may even be orthogonal and thus, may not necessarily correlate well with each other.

The evolution of viruses like HIV under drug and immune selection pressures induces correlated mutations due to constraints on the structural stability and fitness (ability to assemble, replicate, and propagate infection) of the virus [48]. This is a manifestation of the epistatic interactions in the viral genome. The analysis presented here provides a framework based on sequence prevalence to examine the role of correlated mutations in determining the structural and functional fitness landscape of HIV proteins, especially under drug-selection pressure.

Epistatic effects are vital in shaping the higher order (well beyond pairwise) "prevalence" landscape of HIV proteins involved in engendering drug resistance. Identifying/elucidating the epistatic effects for key resistance mutations can help in designing better experiments to probe epistasis and has the potential to impact future HIV drug therapies.

## Materials and methods

The Potts Hamiltonian model of protein sequence covariation is a probabilistic model built from the single and pairwise site amino-acid frequencies in a protein multiple sequence alignment, and aimed at describing the probabilities of observing different sequences in the MSA. To approximate the unknown empirical probability distribution $P(S)$ that best describes a sequence $S$ of length $L$ with each residue encoded in a $Q$-letter alphabet using a model probability distribution $P^m(S)$, we choose the maximum entropy or least biased distribution as the model distribution. Similar distributions that maximize the entropy, with the constraint that the empirical univariate and bivariate marginal distributions are preserved, have been derived in [10, 11, 22, 32, 49]. We follow a derivation of the maximum entropy model in [32, 50], which takes the form of an exponential distribution:

$$E(S) = \sum_i^L h_{S_i}^i + \sum_{i=1}^L \sum_{j=1}^{i-1} J_{S_i S_j}^{ij} \tag{2}$$

$$P^m(S) \propto e^{-E(S)} \tag{3}$$

where the quantity $E(S)$ is the Potts statistical energy of a sequence $S$ of length $L$; the model parameters $h_{S_i}^i$ called "fields" represent the contribution to statistical energy from a residue $S_i$ at position $i$ in $S$, and $J_{S_i S_j}^{ij}$ called "couplings" represent the energy contribution from a pair of residues at positions $i, j$. In this form, the Potts Hamiltonian consists of $LQ$ "field" terms and $\binom{L}{2} Q^2$ "coupling" terms. For the distribution $P^m \propto e^{-E}$, negative fields and couplings indicate favored amino acids. The change in Potts energy for a mutation $\alpha \rightarrow \beta$ at position $i$ in $S$ is given by:

$$\Delta E(S_{\alpha \rightarrow \beta}^i) = E(S_\alpha^i) - E(S_\beta^i) = h_\alpha^i - h_\beta^i + \sum_{j \neq i}^L J_{\alpha S_j}^{ij} - J_{\beta S_j}^{ij} \tag{4}$$

In this form, $\Delta E(S_{\alpha \rightarrow \beta}^i) > 0$ implies that residue $\beta$ is more favorable than residue $\alpha$ at the given position and *vice versa*.

While the original model developed by *Potts et al.* only included nearest neighbor interactions and spin state vectors distributed across a hypersphere [51–53], in contrast, the "spin" models used in biological physics which correspond to global models of protein sequence-covariation, are generalizations of the Potts model with each "spin" state representing the amino-acid character at a given position in the protein, and able to interact with every other "spin" ("infinite range"). Such spin-models have been well established in the maximum entropy protein sequence-covariation literature and are often referred to simply as the Potts or Ising (in case of just two spins or amino-acid residues, wildtype and mutant) model [17, 20–23, 25, 33–35, 37, 38, 54]. In accordance with the literature, we refer to this model concisely as simply the Potts model.

## Data processing

HIV protein multiple sequence alignments for protease, reverse transcriptase, and integrase are obtained from the Stanford University HIV Drug Resistance Database (HIVDB, https://hivdb.stanford.edu) [55, 56] using the genotype-rx search (https://hivdb.stanford.edu/pages/genotype-rx.html) (alternatively, downloadable datasets are also available at https://hivdb.stanford.edu/pages/geno-rx-datasets.html) and filtered using the criteria HIV-1 subtype B and nonCRF, drug-experienced (# of protease inhibitors or PIs = 1–9 for PR, # of nucleoside analog reverse transcriptase inhibitors or NRTIs = 1–9 and # of non-nucleoside analog reverse transcriptase inhibitors or NNRTIs = 1–4 for RT, and # of integrase strand-transfer inhibitors or INSTIs = 1–3 for IN); and we remove sequences with mixtures or ambiguous amino acids. Complete sequences with any insertions ('#') or deletions ('∼') are removed. Such sequences form a small (<1%) fraction of the MSA and removing them doesn't significantly affect the MSA statistics. MSA columns with more than 1% "dots" ('.') which represent unsequenced positions in the sequences are removed to avoid spurious correlations in the subsequent Potts model built on the MSA. Remaining sequences with any "dots" or unsequenced positions are then removed. This resulted in a final MSA size of $N$ = 5710 sequences of length $L$ = 99 for PR, $N$ = 19194 sequences of length $L$ = 188 for RT, and $N$ = 1220 sequences of length $L$ = 263 for IN. For RT, sequences with exposure to both NRTIs and NNRTIs were selected due to much lesser number of sequences exposed to only NRTI or only NNRTI being available. Multiple sequence alignments for the p24 Capsid protein are obtained from the the Los Alamos HIV Sequence Database [57] using the customizable advanced search interface https://www.hiv.lanl.gov/components/sequence/HIV/asearch/map_db.comp and selecting for subtype B and nonCRF, etc. Sequences with inserts/deletions are filtered out. For capsid, drug exposure data and a comprehensive list of drug-resistance mutations are not yet available; drug-naive sequences are used. The subtype B consensus sequence is obtained from the Los Alamos HIV sequence database [57] consensus and ancestral sequence alignments (https://www.hiv.lanl.gov/content/sequence/HIV/CONSENSUS/Consensus.html, last updated August 2004). The subtype B consensus sequence is referred to as the 'consensus/wild-type' throughout the text.

It has been previously established that phylogenetic corrections are not required for HIV patient protein sequences [23, 32] as they exhibit star-like phylogenies [46, 58]. For model inference, HIV patient sequences, are given sequence weights such that the effective number of sequences obtained from any single patient is 1. Sequences obtained from different patients are considered to be independent.

## Mutation information

Drug resistance information, including a list of drug-resistance associated mutations are obtained from the Stanford HIVDB (https://hivdb.stanford.edu/dr-summary/resistance-notes) and from [59]. Mutations in HIV are generally classified into three categories: primary, accessory, and polymorphic. Mutations occurring as natural variants in drug-naive individuals are referred to as polymorphic mutations. Mutations affecting in vitro drug-susceptibility, occurring commonly in patients experiencing virological failure, and with fairly low extent of polymorphism are classified as major or primary drug-resistance mutations. In contrast, mutations with little or no effect on drug susceptibility directly but reducing drug susceptibility or increasing fitness in combination with primary mutations are classified as accessory. For this work, mutations classified as both primary/accessory are considered as drug-resistance associated mutations.

## Alphabet reduction

A reduced grouping of alphabets based on statistical properties can capture most of the information in the full 20 letter amino acid alphabet while decreasing the dimensionality of the parameter space leading to more efficient model inference [17, 19, 22]. All possible alphabet reductions from a $Q$-letter alphabet to a $Q-1$ letter alphabet at a site $i$ are enumerated for all pairs of positions $i, j$ ($j \neq i$) by summing the bivariate marginals for each of the $Q2$ possible combinations and selecting the alphabet grouping that minimizes the root mean square difference (RMSD) in the mutual information (MI):

$$MI_{RMSD} = \sqrt{\frac{1}{N} \sum_{ij} (MI_Q^{ij} - MI_{Q-1}^{ij})^2} \tag{5}$$

The process is then iteratively carried out until the desired reduction in amino acid characters is achieved. Using the reduced alphabet, the original MSA is then re-encoded and the bivariate marginals are recalculated. Small pseudocounts are added to the bivariate marginals, as described in [17, 23, 35] to make up for sampling biases, or to limit divergences in the inference procedure.

Due to residue conservation at many sites in HIV-1, several studies have used a binary alphabet to extract meaningful information from sequences ([32, 60, 61]). A binary alphabet however, marginalizes the information at a site to only the wild-type and mutant residues with the loss of some informative distinctions between residues at sites acquiring multiple mutations. To strike a balance between loss of information and the reduction of the number of degrees of freedom, we choose a reduced alphabet of 4 letters. Our 4 letter alphabet reduction gives a *Pearson's $R^2$* coefficient of 0.995, 0.984, 0.980, and 0.992, for protease, reverse transcriptase, integrase, and p24 capsid protein, respectively between the *MI* of bivariate marginal distributions with the full 21 letter alphabet and the reduced 4 letter alphabet, representing minimal loss of information due to the reduction.

Due to reduction in alphabet, some mutations may not be amenable to our analysis when comparing to experimental fitness measurements such as replicative capacities, etc. We choose mutations corresponding to marginals with higher values in the MSA to be more representative of the model predictions.

## Model inference

The goal of the Potts model inference is to find a suitable set of fields and couplings $\{h, J\}$ parameters that fully determine the Potts Hamiltonian $E(S)$, and best reproduce the empirical bivariate marginals.

A number of techniques have been developed for inferring the model parameters previously [10, 11, 22, 32, 49, 62–66]. The methodology followed here is similar to the one in [32], where, the bivariate marginals are estimated by generating sequences through a Markov Chain Monte Carlo (MCMC) sampling procedure, given a set of fields and couplings. The Metropolis criterion for the generated sequence(s) is proportional to their Potts energies. This is followed by a gradient descent step using a multidimensional Newton search, to determine the optimal set of Potts parameters that minimizes the difference between the empirical bivariate marginal distribution and the bivariate marginal estimates from the MCMC sample. The scheme for choosing the Newton update step is important. A quasi-Newton parameter update approach determining the updates to $J_{S_i S_j}^{ij}$ and $h_{S_i}^i$ by inverting the system's Jacobian was developed in [32], which we follow here. Although the methodology involves approximations during the computation of the Newton steps, the advantage of the methodology is that it avoids making

explicit approximations to the model probability distribution at the cost of being heavily computationally intensive. We have employed a GPU implementation of the MCMC methodology, which makes it computationally tractable without resorting to more approximate inverse inference methods. The MCMC algorithm implemented on GPUs has been previously used to infer accurate Potts models in [17, 23, 35, 38, 54, 67].

The computational cost of fitting $\binom{L}{2} * (4-1)^2 + L * (4-1)$ model parameters for the smallest protein in our analysis, PR, on 2 NVIDIA K80 or 4 NVIDIA TitanX GPUs is $\approx 20h$. For a more detailed description of data preprocessing, model inference, and comparison with other methods, we refer the reader to [19] and [17, 23, 38, 54]. A repository containing the final MSAs is available at https://github.com/ComputationalBiophysicsCollaborative/HIV_MSAs.

The site-independent model is inferred based on the univariate marginals, or the residue frequencies, $f_\alpha^i$ (for a residue $\alpha$ at position $i$) in the MSA alone, giving the "field" parameters as:

$$h_\alpha^i = -\ln f_\alpha^i \tag{6}$$

A small pseudo-count is added to the $f_\alpha^i$ to avoid indeterminate logarithms. The independent model energies of a sequence $S$ are given as $E(S) = \sum_i^L h_{S_i}^i$, where $i$ is a position in the sequence, and $L$ is the length of the sequence.

## Prediction of higher order marginals

The Potts model inferred using the methodology described above is generative, allowing for generation of new synthetic sequences which very closely represent the sequences in the MSA of protein sequences obtained from HIV patients. For prediction and comparison of the higher-order marginals, both the Potts and independent models are used to generate new sequences, and subsequence frequencies (marginals) are compared between the dataset MSA and the Potts/independent model generated MSAs. For each subsequence of length 2–14, the process is repeated for 500 randomly picked subsequences and the Spearman correlation coefficient is calculated for all subsequences which appear with a frequency greater than the threshold (to avoid noise).

## Statistical robustness of HIV Potts models

Finite sampling and overfitting can affect all inference problems, and the inverse Ising inference is no exception. In case of the Potts model, the number of model parameters can vastly outsize the number of sequences in the MSA, yet it is possible to fit accurate Potts models [54] to those MSAs, as the model is not directly fit to the sequences but to the bivariate marginals of the MSA. However, finite sampling can affect the estimation of the marginal distributions, which, in turn, affects model inference. In fact, overfitting in the inverse Ising inference arises due to the finite-sampling error in the bivariate marginals estimated from a finite-sized MSA. The degree of overfitting can be quantified using the "signal-to-noise ratio" (SNR), which is a function of the sequence length $L$, alphabet size $q$, number of sequences in the MSA $N$, and the degree of evolutionary conservation in the protein. The SNR for Potts models fit to protein sequences is discussed in more detail in [54]. If the SNR is small, the Potts model is unable to reliably distinguish high scoring sequences in the data set from low-scoring sequences. If SNR is close to or greater than 1, then overfitting is minimal and the Potts model is more reliable. In the analysis presented here, IN has the lowest SNR (0.14 compared to 43.7 for RT, and 21.6 for PR) on account of being one of the more conserved proteins with the lowest number of sequences in the MSA, and may be more affected by overfitting. Different predictions of the

Potts model, however are differently affected by finite sampling errors with predictions of $\Delta E$s which form the basis of Potts model "fitness" predictions among the more robust [54]. The Potts model is also able to accurately capture the higher-order marginals in the MSA. Thus, we conclude that the MSA sample sizes used in this study are sufficiently large to construct Potts models for these HIV proteins that adequately reflect the effect of the sequence background on mutations.

### Protein stability analysis

The changes in folding free energies due to mutations are analyzed using FoldX [44, 45], which uses an empirical force field to determine the energetic effects of point mutations. The HIV-1 protease structure (PDB: 3S85) is used as reference, repaired using the RepairPDB function in the FoldX suite, and the free energy of mutants is calculated with the BuildModel function under default parameters. For each mutation, the mean of 10 FoldX calculations is used as the $\Delta\Delta G$ value.

### Supporting information

**S1 File. Supplementary methods, details and figures.** Details of the evolutionary conservation in different HIV enzymatic proteins and its effect on the observable evidence for epistasis is given in Section 1. Details on comparison with different experimental measures of fitness and Potts and independent models, along with comparisons with FoldX predicted changes in folding energetics due to mutations are given in Section 2, as well as details of why comparisons for double mutant cycles are difficult. Details of the weak evidence for epistasis that can be drawn from hamming distance distributions are given in Section 3.
(PDF)

**S2 File. Supplementary figures and tables for double mutant cycles.** Figures and tables showing the distribution of Potts model predicted double mutant cycle effects for all double mutations indicating the strongest, predicted double mutant cycle effects involving mutations (at least one amongst the pair) at drug-resistance-associated sites and corresponding literature references in HIV-1 protease and integrase are given.
(PDF)

### Author Contributions

**Conceptualization:** Avik Biswas, Allan Haldane, Ronald M. Levy.

**Data curation:** Avik Biswas, Allan Haldane.

**Formal analysis:** Avik Biswas, Allan Haldane, Ronald M. Levy.

**Funding acquisition:** Ronald M. Levy.

**Investigation:** Avik Biswas, Allan Haldane, Ronald M. Levy.

**Methodology:** Avik Biswas, Allan Haldane, Ronald M. Levy.

**Visualization:** Avik Biswas, Allan Haldane, Ronald M. Levy.

**Writing – original draft:** Avik Biswas, Allan Haldane, Ronald M. Levy.

**Writing – review & editing:** Avik Biswas, Allan Haldane, Ronald M. Levy.

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
