## [Decision Letter · Decision Letter 0]

29 Nov 2021

PONE-D-21-33103Limits to detecting epistasis in the fitness landscape of HIVPLOS ONE

Dear Dr. Levy,

Thank you for submitting your manuscript to PLOS ONE. After careful consideration, we feel that it has merit but does not fully meet PLOS ONE’s publication criteria as it currently stands. Therefore, we invite you to submit a revised version of the manuscript that addresses the points raised during the review process. Please submit a letter and a revised manuscript that addresses the minor points raised by the third reviewer. Disregard the comment concerning the quality of the figures. The low resolution of the figures in the prepared pdf is an artifact of the editorial process.

We look forward to receiving your revised manuscript.

Kind regards,

Emilio Gallicchio, Ph.D.

Academic Editor

PLOS ONE

Journal Requirements:

"We thank Chu’nan Liu for help with the FoldX suite and Vincenzo Carnevale for his helpful comments and suggestions. We also thank the very supportive and collaborative environment provided by the HIV Interaction and Viral Evolution (HIVE) Center at the Scripps Research Institute (http://hivescripps.edu). This work has been supported by the National Institutes of Health grants U54-GM103368, R35-GM132090, and S10OD020095, and the National Science Foundation grant 1934848. The authors declare no competing interests. "

"This work has been supported by the National Institute of General Medical Science (www.nigms.nih.gov) and the National Institute of Allergy and Infectious Diseases (www.niaid.nih.gov) of the National Institutes of Health (www.nih.gov) grants U54-GM103368 and R35-GM132090 to R.M.L, the National Institutes of Health (www.nih.gov) grant S10OD020095 to R.M.L and the National Science Foundation (www.nsf.gov) grant 1934848 to R.M.L and A.H. The funders had no role in study design, data collection and analysis, decision to publish, or preparation of the manuscript."

4. PLOS requires an ORCID iD for the corresponding author in Editorial Manager on papers submitted after December 6th, 2016. Please ensure that you have an ORCID iD and that it is validated in Editorial Manager. To do this, go to ‘Update my Information’ (in the upper left-hand corner of the main menu), and click on the Fetch/Validate link next to the ORCID field. This will take you to the ORCID site and allow you to create a new iD or authenticate a pre-existing iD in Editorial Manager. Please see the following video for instructions on linking an ORCID iD to your Editorial Manager account: https://www.youtube.com/watch?v=_xcclfuvtx

Reviewers' comments:

Reviewer's Responses to Questions

**Comments to the Author**

1. Is the manuscript technically sound, and do the data support the conclusions?

Reviewer #1: Yes

2. Has the statistical analysis been performed appropriately and rigorously? 

Reviewer #1: Yes

3. Have the authors made all data underlying the findings in their manuscript fully available?

Reviewer #1: Yes

4. Is the manuscript presented in an intelligible fashion and written in standard English?

Reviewer #1: Yes

5. Review Comments to the Author

Reviewer #1: The work of Biswas et al. focuses on the use of data-driven models of protein sequence composition to characterize the fitness landscape of proteins, particularly HIV proteins that are commonly targeted by antiretroviral drugs. This work deals with a common and relevant problem in the field of amino acid coevolution where both sequence data (or its diversity) is scarce, and where experimental measures of fitness are limited by experimental constraints. This extensive study has two important results: 1) higher order marginals can be recovered with a pairwise model and that such higher order marginals indeed play a role in the fitness landscape of viral proteins. 2) It sheds light on why existent experimental data trying to capture epistatic effects of mutations is hard to model due to the limited dynamic range of these experiments.

Being this a revision of an initial submission, I was able to review previous reviewers comments and the accompanying responses and changes. My assessment is that the responses to the reviewer comments were appropriate and the changes did in fact improve the clarify of the presentation. My general assessment from the previous reviewers was that they did not have enough background in the recent developments in the field of amino acid coevolution and were focused on challenging the model development itself which has been already established and refined in the literature. This revision is technically sound but also does a good job presenting the biological implications of their results. In conclusion, I think this study is relevant for the field and explains several open questions concerning the application of sequence Potts models to viral proteins. I find this study relevant and of use, especially in these times where understanding the fitness landscapes of proteins related to infectious diseases in a matter of pressing public health.

Given the extensive changes toward this revision, I only have a few questions and comments that if responded could make this manuscript more clear.

General

1. The finding that the Capsid shows a higher correlation with he Potts model is interesting. A similar correlation has been observed for Capsids on other types of viruses (AAV) providing further evidence of the relevance of Potts models in these types of studies. I suggest citing this additional study too (https://doi.org/10.1016/j.bpj.2020.12.018).

2. Data Processing. It is stated that positions with more than 1% gaps of gaps were removed. This seems to me like a very stringent cutoff, can the authors provide an explanation of why more than 1% is too much?

3. Data processing. The statement “Sequences with insertions and deletions are removed” is not clear to me. Do you mean the positions with insertions and deletions? Or are you removing any complete sequence that has an insertion or deletion? Somehow this does not makes sense to me.

4. Alphabet reduction. I wonder if Equation 5 could be improved by including an index instead of the explicit realizations like Q=20, my understanding is that after each iteractive step, Q will be reduced with respect to the previous step right? If this is true then it should be noted that this is not only valid for a change from 21 to 20. If I misinterpretted it, then it is possible that the equation needs some further clarification.

5.Alphabet reduction. The authors compared the model with alphabet reduction with the model using full parameters. If the model with the full parameters was inferred, then what was the motivation to reduce the alphabet?

6. In the section statistical robustness of HIV Potts models, the concept of Signal-to-noise ratio is introduced and scores for the different protein systems are presented. Although it is mentioned that SNR depends on several factors, it would be good to have a more concrete definition or point out to previous work where it is defined.

7. The quality of some images is quite low, I assume this is due to the system that converts the manuscript for review. Please make sure to have high quality images. Figure 3 axes could have larger fonts for readability.

Minor

1. The use of the term favorability/ disfavorability is a bit awkward, I would consider another term.

2. Conventions, should “Fig 1” be spelled “Fig. 1” ?

3. Page 5, line 149. Change “.. higher entropies in Supplementary File 1 Fig 2 A” to “.. higher entropies (Supplementary File 1 Fig 2 A).”

4. Page 5, line 157. Change “This is suggestive that strong couplings ..” to “This is suggestive of strong couplings”

5. Page 11, line 364. correct “changes in CA perhaps has a ..” to “changes in CA perhaps have a ..”

6. NRTIs is only defined in a figure caption, I suggest to define it also in the main text.

7. Data processing. The concept of “deletes” is used instead of “deletion”, I would simply use deletions.

8. Page 13, line 444. Add a period after “filtered out.”

9. Page 13, correct “drug resistance mutations is not yet” with “drug resistance mutations are not yet”

10. Mutation information section. Replace “including list of ..” with “ including a list of ..”

11. Subsection title. Use lower case across titles, e.g. Change “Statistical Robustness of HIV” with “Statistical robustness of HIV ..”

12. Change title in the SI to be compatible with the manuscript

13. What is the need of having two distinct Supplementary files? It seems to me that the two files can be combined into a single document

6. PLOS authors have the option to publish the peer review history of their article (what does this mean?). If published, this will include your full peer review and any attached files.

Reviewer #1: No

---

## [Author Response · Author response to Decision Letter 0]

18 Dec 2021

In response to the specific editor comments, we have provided a rebuttal letter that responds to each point raised by the academic editor and reviewer(s), a marked-up copy of the manuscript highlighting changes made to the original version, as well as an unmarked version of the revised paper without tracked changes. 

In response to the specific comments raised by the reviewer, we provide a detailed response (in the rebuttal letter) as below:

The work of Biswas et al. focuses on the use of data-driven models of protein sequence composition to characterize the fitness landscape of proteins, particularly HIV proteins that are commonly targeted by antiretroviral drugs. This work deals with a common and relevant problem in the field of amino acid coevolution where both sequence data (or its diversity) is scarce, and where experimental measures of fitness are limited by experimental constraints. This extensive study has two important results: 1) higher order marginals can be recovered with a pairwise model and that such higher order marginals indeed play a role in the fitness landscape of viral proteins. 2)It sheds light on why existent experimental data trying to capture epistatic effects of mutations is hard to model due to the limited dynamic range of these experiments.

Being this a revision of an initial submission, I was able to review previous reviewers comments and the accompanying responses and changes. My assessment is that the responses to the reviewer comments were appropriate and the changes did in fact improve the clarify of the presentation. My general assessment from the previous reviewers was that they did not have enough background in the recent developments in the field of amino acid coevolution and were focused on challenging the model development itself which has been already established and refined in the literature. This revision is technically sound but also does a good job presenting the biological implications of their results. In conclusion, I think this study is relevant for the field and explains several open questions concerning the application of sequence Potts models to viral proteins. I find this study relevant and of use, especially in these times where understanding the fitness landscapes of proteins related to infectious diseases in a matter of pressing public health.

Given the extensive changes toward this revision, I only have a few questions and comments that if responded could make this manuscript more clear.

We thank the reviewer for the thoughtful commentary and for highlighting the main points of the manuscript. The reviewer finds the study "relevant and of use". Below we provide a point-by-point response to the questions and comments raised by the reviewer. We have also updated the manuscript with changes made in response to the current reviewer comments highlighted in "cyan", while changes made to the previous reviewers' comments are highlighted in "yellow". 

General:

1. The finding that the Capsid shows a higher correlation with the Potts model is interesting. A similar correlation has been observed for Capsids on other types of viruses (AAV) providing further evidence of the relevance of Potts models in these types of studies. I suggest citing this additional study too (https://doi.org/10.1016/j.bpj.2020.12.018).

We thank the reviewer for the suggestion and have added the citation in the updated manuscript.

2. Data Processing. It is stated that positions with more than 1% gaps were removed. This seems to me like a very stringent cutoff, can the authors provide an explanation of why more than1% is too much?

We had not used the correct wording, "gap"; whiIe referring to the "dot" character in HIV protein sequence alignments available from the Stanford HIV drug resistance database, which represents an unsequenced position in the sequence. Such positions (columns in the multiple sequence alignment) with more than 1% "dots" or missing amino characters were removed from the MSA, so that the subsequent Potts model built on the MSA would not have spurious correlations between missing amino acids/unsequenced positions and amino acid residues at other positions. Similarly remaining sequences (rows in the MSA) with "dots" (a small fraction of the MSA) after the first filtration step, were then removed to preserve the quality of sequences. We have made this clear by revising the previous statement in the Materials and Methods section: 

" MSA columns with more than 1% ``dots" ('.') which represent unsequenced positions in the sequences are removed to avoid spurious correlations in the subsequent Potts model built on the MSA. Remaining sequences with any "dots" or unsequenced positions are then removed."

3. Data processing. The statement “Sequences with insertions and deletions are removed” is not clear to me. Do you mean the positions with insertions and deletions? Or are you removing any complete sequence that has an insertion or deletion? Somehow this does not makes sense to me.

Complete sequences with insertions or deletions are removed. There are only few such sequences and removing them doesn't affect the multiple sequence alignment (MSA) statistics much; for example, less than 1% (~0.3%) of the sequences in the Reverse Transcriptase MSA contain insertions or deletions. On the other hand, keeping these sequences in the MSA would have complicated the subsequent model building, without improving the statistics. We have now made this clear in the revised text: 

" Complete sequences with any insertions ('#') or deletions ('~') are removed. Such sequences form a small fraction (<1%) of the MSA and removing them doesn't significantly affect the MSA statistics. "

4. Alphabet reduction. I wonder if Equation 5 could be improved by including an index instead of the explicit realizations like Q=20, my understanding is that after each iterative step, Q will be reduced with respect to the previous step right? If this is true then it should be noted that this is not only valid for a change from 21 to 20. If I misinterpreted it, then it is possible that the equation needs some further clarification.

The reviewer is right. In each iterative step, the alphabet is reduced by 1 from a Q-letter alphabet in the previous step to a Q-1 letter alphabet. To make this clear, we have modified Equation 5 with an index instead of explicit realizations as below:

〖MI〗_RMSD= √(1/N ∑_ij▒(〖MI〗_Q^ij-〖MI〗_(Q-1)^ij )^2 )

5.Alphabet reduction. The authors compared the model with alphabet reduction with the model using full parameters. If the model with the full parameters was inferred, then what was the motivation to reduce the alphabet?

We apologize for the confusion. We only compared the Mutual Information (MI) of the MSA encoded in the full 21-letter (20 amino acids + 1 gap character) alphabet to the MI of the MSA in a reduced alphabet. The Potts model is only inferred based on the MSA in the reduced alphabet for computational efficiency and not inferred based on the MSA encoded in the full 21-letter alphabet. In the most recent work in our group, we are using the full alphabet. The project described in this manuscript was carried out using a reduced alphabet. We do not believe the alphabet reduction affects any of the results presented in this manuscript.

6. In the section statistical robustness of HIV Potts models, the concept of Signal-to-noise ratio is introduced and scores for the different protein systems are presented. Although it is mentioned that SNR depends on several factors, it would be good to have a more concrete definition or point out to previous work where it is defined.

We have now included a reference to previous work from the group which gives a more elaborate definition of the SNR and the several factors it depends on:

" The SNR for Potts models fit to protein sequences is discussed in more detail in [54]. "

Minor:

1. The use of the term favorability/ disfavorability is a bit awkward, I would consider another term.

We have used the terms "favorability/disfavorability" for a mutation in its specific sequence background, in keeping with previous published work (Biswas et al., eLife, 2019). The Potts model describes the prevalence landscapes of the protein sequences, and the model �Es best describe the favorability/disfavorability of mutations in a given sequence background. Alternatively, using terms like stabilizing/destabilizing (Flynn et al., MBE, 2017) can lead to further confusion as they can be interpreted to be pertaining specifically to protein stabilities.

2. Conventions, should “Fig 1” be spelled “Fig. 1” ?

We have now renamed all figures in the manuscript according to the convention, for example Fig 1 as Fig. 1.

3. Page 5, line 149. Change “.. higher entropies in Supplementary File 1 Fig 2 A” to “.. higher entropies (Supplementary File 1 Fig 2 A).”

We have modified the sentence in the Main text.

4. Page 5, line 157. Change “This is suggestive that strong couplings ..” to “This is suggestive of strong couplings”

We thank the reviewer for pointing out the typo and have corrected it in the revised text.

5. Page 11, line 364. correct “changes in CA perhaps has a ..” to “changes in CA perhaps have a ..”

We have corrected the typo in the revised text.

6. NRTIs is only defined in a figure caption, I suggest to define it also in the main text.

We thank the reviewer for pointing this out and have now defined all the major drug-classes used in antiretroviral therapy including NRTIs in the Main text lines 432-436.

7. Data processing. The concept of “deletes” is used instead of “deletion”, I would simply use deletions.

We have changed the word "deletes" to "deletions".

8. Page 13, line 444. Add a period after “filtered out.”

Added a period.

9. Page 13, correct “drug resistance mutations is not yet” with “drug resistance mutations are not yet”

Corrected.

10. Mutation information section. Replace “including list of ..” with “ including a list of ..”

We have modified the main text replacing "including list of .." with "including a list of .." as pointed out by the reviewer.

11. Subsection title. Use lower case across titles, e.g. Change “Statistical Robustness of HIV” with “Statistical robustness of HIV ..”

We have now modified the font for the subsection titles accordingly.

12. Change title in the SI to be compatible with the manuscript. 

We have now changed the title of the SI to be compatible with the manuscript. 

13. What is the need of having two distinct Supplementary files? It seems to me that the two files can be combined into a single document.

The supplementary files are kept separate for ease of readability. Supplementary file 1 contains the details of many methods, supplementary data, and figures. Supplementary file 2 on the other hand contains only supplementary tables and figures pertaining to double mutant cycles showing that most strongly coupled pairs of mutations predicted by the Potts model contain drug-resistance associated mutations studied in the literature.

Note that there was a slight error in Figure S2A in Supplementary File 2, in marking the standard deviations, which we have now corrected and modified in the revised version. This does not affect any of the results.

---

## [Editor Report · Decision Letter 1]

21 Dec 2021

Limits to detecting epistasis in the fitness landscape of HIV

PONE-D-21-33103R1

Dear Dr. Levy,

We’re pleased to inform you that your manuscript has been judged scientifically suitable for publication and will be formally accepted for publication once it meets all outstanding technical requirements.

Kind regards,

Emilio Gallicchio, Ph.D.

Academic Editor

PLOS ONE
---

## [Editor Report · Acceptance letter]

6 Jan 2022

PONE-D-21-33103R1 

Limits to detecting epistasis in the fitness landscape of HIV 

Dear Dr. Levy:

I'm pleased to inform you that your manuscript has been deemed suitable for publication in PLOS ONE. Congratulations! Your manuscript is now with our production department. 

Kind regards, 

on behalf of

Dr Emilio Gallicchio 

Academic Editor

PLOS ONE